# Installation of synergistic binding sites onto porous organic polymers for efficient removal of perfluorooctanoic acid

Xiongli Liu[1], Changjia Zhu[2], Jun Yin [3,4], Jixin Li[5], Zhiyuan Zhang[1], Jinli Li[1], Feng Shui[1], Zifeng You[1], Zhan Shi [5], Baiyan Li [1✉], Xian-He Bu [1✉], Ayman Nafady[6] & Shengqian Ma [2✉]

Herein, we report a strategy to construct highly efficient perfluorooctanoic acid (PFOA) adsorbents by installing synergistic electrostatic/hydrophobic sites onto porous organic polymers (POPs). The constructed model material of PAF-1-NDMB (NDMB = N,N-dimethyl-butylamine) demonstrates an exceptionally high PFOA uptake capacity over 2000 mg g$^{-1}$, which is 14.8 times enhancement compared with its parent material of PAF-1. And it is 32.0 and 24.1 times higher than benchmark materials of DFB-CDP (β-cyclodextrin (β-CD)-based polymer network) and activated carbon under the same conditions. Furthermore, PAF-1-NDMB exhibits the highest $k_2$ value of 24,000 g mg$^{-1}$ h$^{-1}$ among all reported PFOA sorbents. And it can remove 99.99% PFOA from 1000 ppb to <70 ppt within 2 min, which is lower than the advisory level of Environmental Protection Agency of United States. This work thus not only provides a generic approach for constructing PFOA adsorbents, but also develops POPs as a platform for PFOA capture.

[1] School of Materials Science and Engineering, National Institute for Advanced Materials, TKL of Metal and Molecule-Based Material Chemistry, Nankai University, Tianjin 300350, P. R. China. [2] Department of Chemistry, University of North Texas 1508W Mulberry St, Denton, TX 76201, USA. [3] Physical Sciences and Engineering Division, King Abdullah University of Science and Technology, Advanced Membranes and Porous Materials Center, Thuwal 23955-6900, Kingdom of Saudi Arabia. [4] Kingdom of Saudi Arabia; KAUST Catalysis Center, Physical Sciences and Engineering Division, King Abdullah University of Science and Technology, Thuwal 23955-6900, Kingdom of Saudi Arabia. [5] State Key Laboratory of Inorganic Synthesis and Preparative Chemistry, College of Chemistry, Jilin University, Changchun 130012, P. R. China. [6] Department of Chemistry, College of Science, King Saud University, Riyadh 11451, Saudi Arabia. ✉email: libaiyan@nankai.edu.cn; buxh@nankai.edu.cn; Shengqian.Ma@unt.edu

Per-and poly-fluoroalkyl substances (PFAS), as environmental pollutants, often show an adverse and severe impacts on environmental and human health due to their intrinsic high toxicity, extraordinary prevalence and persistence, and instant bioaccumulation[1–4]. According to the U.S. Environmental Protection Agency (EPA), the PFAS concentration for combined perfluorooctanoic acid (PFOA) and perfluorooctane sulfonate (PFOS) in drinking water should not exceed the healthy advisory level of 70 ppt[5]. Therefore, it is an urgent task to develop advanced technologies for PFAS removal from contaminated water.

Among various technologies used for PFAS removal from contaminated water such as oxidative[6], UV irradiation[7], sonochemical[8], and electrochemical methods[9], adsorption is highly favored owing to its simplicity and high efficiency in the purification process of contaminated water[10]. Thus, a wide range of adsorbents have been investigated for PFAS elimination from polluted waters. Conventional PFAS adsorbents mainly include activated carbon (AC)[11], ion exchange resins[12], minerals[13], molecularly imprinted polymer (MIP)[14], biosorbents[15–18], carbon nanotubes (CNTs)[19], metal-organic frameworks (MOFs)[20], and covalent organic frameworks (COFs)[21]. However, these adsorbents often have some limitations including low capacity ($0.63 \times 10^{-3}$–753 mg g$^{-1}$)[12–14,16,20], long equilibrium time (1–48 h)[11–14,20], weak binding affinity, poor water/chemical stabilities, and low natural organic matter (NOM) selectivity, which limit their practical applications in the treatment of contaminated water[22,23].

To overcome these drawbacks, it is essential to develop ideal types of PFAS-based adsorbents that possess the following features: (i) high hydrophilicity of the adsorbent particles and densely accessible functionalized capturing sites, benefiting for the well water wettability to facilitate the mass transfer of PFOA toward adsorbents in aqueous solutions and thereby achieving high PFAS uptake capacity in water; (ii) strong interactions with PFAS that can accelerate the adsorption rate; (iii) suitable pore size to avoid the co-adsorption of natural organic matter (e.g. humic acid, HA) in contaminated water during the adsorption process; (iv) exceptional water/chemical stability to facilitate regeneration/recyclability.

Considering these factors, we intend to employ porous organic polymers (POPs)[24,25], with porous aromatic framework (PAF) as the representative[26], as a platform to construct PFAS adsorbent due to their advantageous features of ultrahigh surface areas (>5600 m$^2$ g$^{-1}$), adjustable framework structures, tunable pore sizes, readily functionalized pore wall, and exceptional water/chemical stabilities[27–29]. These POPs have been previously applied in the fields of gas storage and separation[30,31], proton conduction[32,33], catalysis[34,35], energy storage[36,37], enzymatic immobilization[38,39], and water treatment[40,41]. Herein, we describe a synergistic binding sites protocol to construct PFAS adsorbent by simultaneously introducing electrostatic and hydrophobic sites onto POPs as adsorption sites (Fig. 1). We hypothesized that such POPs with multi-functionalized sites can synergistically and efficiently bind PFAS molecules with the features of negative charges and hydrophobic chains at the molecular level[42], thus improving the adsorption performance of PFAS sorbents.

As a proof of concept, we report in this contribution a superior PFAS adsorbent by appending quaternary ammonium groups with long hydrophobic chains [e.g. N,N-dimethylpropylamine (NDMP), N,N-dimethyl-butylamine (NDMB), N,N-dimethylhexylamine (NDMH)] onto the pore wall of PAF-1 to obtain PAF-1-NDMP, PAF-1-NDMB, and PAF-1-NDMH, respectively (Fig. 2). The constructed adsorbents contained both positive charges and hydrophobic chains located on an adjacent position,

which can serve as synergistic binding sites to bind the negative charged sites and hydrophobic chains of PFAS via electrostatic and hydrophobic interactions at the molecular level (Fig. 3). As a result, PAF-1-NDMB as an optimized adsorbent exhibits a record high saturation PFOA (a model pollutant among PFAS) uptake capacity of over 2000 mg g$^{-1}$, which is 14.8 times enhancement compared with its parent material of PAF-1. And this value is 32.0 and 24.1 times higher than that of benchmark materials of β-cyclodextrin (β-CD)-based polymer network (DFB-CDP)[16] and coconut shell activated carbon (AC) under same conditions. In addition, PAF-1-NDMB exhibits extremely fast kinetics with the highest $k_2$ value of 24000 g mg$^{-1}$ h$^{-1}$ among all reported sorbent materials. These results thus rank PAF-1-NDMB as the top candidate among all benchmarked PFOA sorbents.

## Results

**Synthesis and characterization.** In this context, PAF-1-NDMP, PAF-1-NDMB, and PAF-1-NDMH were synthesized via chloromethylation of PAF-1 (based on the consideration of the high reactivity of chloromethyl groups with amines)[43,44], which was then treated with N,N-dimethylpropylamine (NDMP), N,N-dimethyl-butylamine (NDMB), N,N-dimethylhexylamine (NDMH) in ethanol, respectively. FT-IR spectra of dehydrated PAF-1-NDMP, PAF-1-NDMB, and PAF-1-NDMH showed new IR bands at 2352, 2868, and 2952 (broad) cm$^{-1}$, which are assigned to the CH$_3$-N$^+$, CH$_2$, and CH$_3$[45,46]. The new IR bands at 1631, 1676, and 1728 (broad) cm$^{-1}$ can be assigned to the C-N of PAF-1-NDMP, PAF-1-NDMB, and PAF-1-NDMH, respectively (Fig. 4a)[45,46]. The successful incorporation of NDMP, NDMB, and NDMH was also proved by XPS spectroscopy analysis (Supplementary Fig. 1), as indicated by the presentation of N signal at a binding energy of 401 eV[47]. Solid-state $^{13}$C NMR studies identify a new broad resonance at 29.8, 31.5, and 29.6 ppm ascribed to NDMP, NDMB, and NDMH carbon atoms, thereby suggesting the successful attachment of NDMP, NDMB, and NDMH onto the phenyl rings of PAF-1 (Fig. 4b). Results obtained from nitrogen elemental analysis revealed that the overall reaction afforded approximately 0.63 NDMP, 0.79 NDMB, and 0.66 NDMH per biphenyl for functionalized PAF-1 series. The nitrogen content (0.00218, 0.00274 and 0.0023 mol g$^{-1}$) is similar to the chloride content (0.00209, 0.00268 and 0.00221 mol g$^{-1}$) in PAF-1-NDMP, PAF-1-NDMB, and PAF-1-NDMH after functionalized modification, which suggests the completed conversion of CH$_2$Cl into Cl$^-$ due to a large excess of NDMP, NDMB, and NDMH reactant in the reaction (Supplementary Table 1). Moreover, scanning electron microscope (SEM) images indicated that these adsorbents exhibit spherical morphology (Supplementary Fig. 2). Moreover, the data obtained from SEM histogram and dynamic laser light scattering (DLS) suggested that PAF-1-NDMP, PAF-1-NDMB, and PAF-1-NDMH possess similar particle size distribution (0.75–2.55 μm) (Supplementary Figs. 3 and 4). This means that there is no direct correlation between particle sizes and adsorption performances in the following sorption studies. Energy-dispersive X-ray spectroscopy (EDS) maps show the uniform distribution of nitrogen elementary in PAF-1-NDMP, PAF-1-NDMB, and PAF-1-NDMH (Supplementary Fig. 5). N$_2$ adsorption isotherms recorded at 77 K showed that the grafting of NDMP, NDMB and NDMH groups resulted in a massive decrease in the Brunauer-Emmett-Teller (BET) surface area from 3568 m$^2$ g$^{-1}$ for PAF-1 to 602, 108, and 50 m$^2$ g$^{-1}$ for PAF-1-NDMP, PAF-1-NDMB, and PAF-1-NDMH, respectively (Supplementary Fig. 6). Pore size distributions of functionalized PAF-1 indicates the existence of the large pores with the pore sizes of 12.6 Å, 22.8 Å, and 21.6 Å (Supplementary Fig. 7a) for PAF-1-NDMP, PAF-1-NDMB, and PAF-1-NDMH,

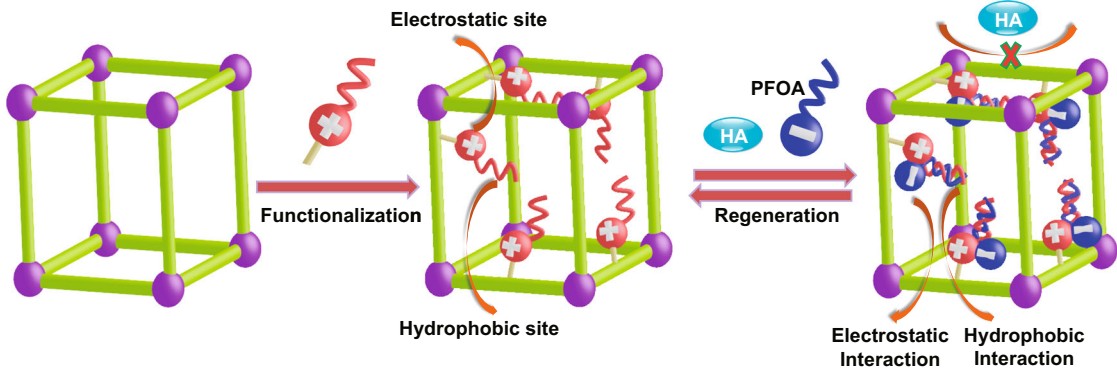

**Fig. 1 'Synergistic binding sites' strategy for PFOA removal.** Illustration of 'Synergistic binding sites' strategy to construct highly efficient sorbents for PFOA removal. (HA represents humic acid, a model compound of organic co-contaminants or natural organic matters).

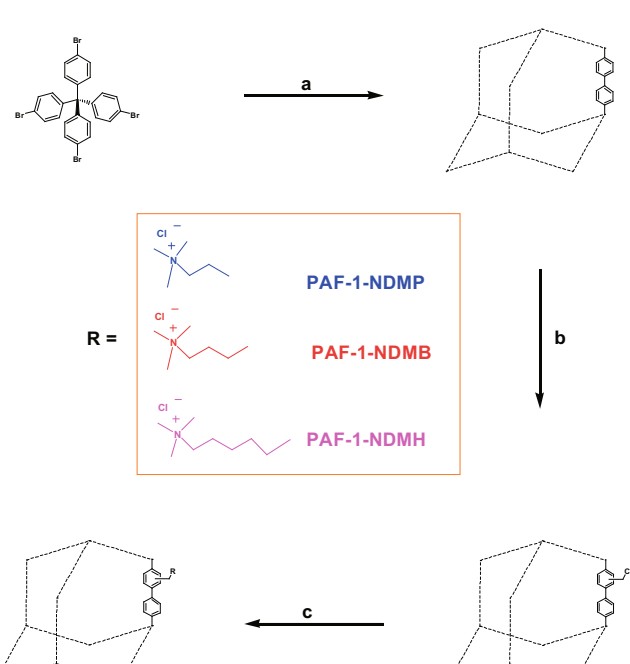

**Fig. 2 General procedure for the synthesis of PFOA adsorbents including PAF-1-NDMP, PAF-1-NDMB, PAF-1-NDMH.** Reaction conditions: **a** Bis(1,5-cyclooctadiene)nickel(0), 1,5-cyclooctadiene, 2,2'-bipyridine, N,N-dimethylformamide, and tetrahydrofuran, room temperature. **b** Paraformaldehyde, AcOH, H₃PO₄, HCl, 90 °C. and **c** N,N-dimethylpropylamine (NDMP), N,N-dimethyl-butylamine (NDMB), or N,N-dimethylhexylamine (NDMH) in EtOH, 90 °C.

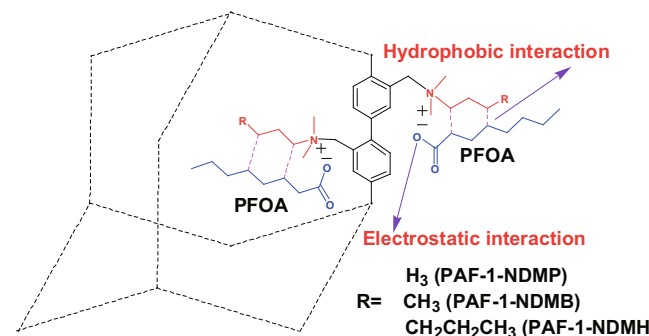

**Fig. 3 Illustration of interactions of functionalized PAF-1 with PFOA.** Electrostatic sites and hydrophobic sites for PFOA adsorption in functionalized PAF-1 with both positively charged sites and hydrophobic sites.

respectively, which are large enough to allow the entrance of PFOA molecules with dimension of $11.0 \times 3.6 \times 3.6$ Å (Supplementary Fig. 7b). In addition, after introducing the quaternary ammonium groups onto PAF-1, it results in a hydrophobic pore surface for PAF-1 particles with water contact angle of 116.6° to a hydrophilic pore surface for adsorbent particles with water contact angles of 38.2°, 34.9°, and 36.3° for PAF-1-NDMP, PAF-1-NDMB, and PAF-1-NDMH, respectively (Supplementary Fig. 8). High hydrophilicity of these adsorbent particles favors for the improvement of adsorptive performance toward PFOA.

**PFOA sorption studies.** To assess the performance of the functionalized PAF-1 materials as sorbents to extract PFOA from aqueous solutions, the as-synthesized materials (20.0 mg) were put into a 50 mL aqueous solution (pH = 6.88) with PFOA

concentration of 1000 ppb at room temperature (see methods for details). A Cl⁻ content of 0.0018, 0.00227 and 0.00185 mol g⁻¹ was observed in filtrate of PAF-1-NDMP, PAF-1-NDMB, and PAF-1-NDMH after PFOA adsorption, which indicated ion exchange (electrostatic interactions mechanism), occurred during PFOA adsorption process (Supplementary Table 2). Moreover, the Cl⁻ content of PFOA loaded functionalized adsorbents including PFOA@PAF-1-NDMP, PFOA@PAF-1-NDMB, and PFOA@PAF-1-NDMH are 0.00028, 0.00040, and 0.00035 mol g⁻¹, respectively (Supplementary Table 3), much lower than pristine PAF-1-NDMP, PAF-1-NDMB, and PAF-1-NDMH, which further verified the ion exchange mechanism. The BET surfaces and pore sizes of all as-prepared materials decreased after the loading of PFOA, which could be ascribed to PFOA molecules filled into the pore of absorbents (Supplementary Table 4, Supplementary Figs. 9 and 10). In addition, the contact angle increased after adsorption process, which also suggested the loading of PFOA molecules inside the nanopores of adsorbents (Supplementary Table 4, Supplementary Fig. 11). The sorption results showed that PAF-1-NDMB exhibit an optimized sorption performance among these as-prepared materials (Supplementary Figs. 12–15). Hence, PAF-1-NDMB was selected as a model sorbent for further investigation. As shown in Fig. 5a, remarkably fast kinetics was noted for PAF-1-NDMB, consistent with a 99.99% removal of PFOA within 2 min, together with a huge reduction in PFOA concentration from 1000 ppb to 54 ppt. This value is lower than that of the health advisory level (70 ppt) for drinking water by EPA of United States[5]. While only the removal of 35.8%, 80%, and 60.4% were observed at equilibrium time of approximately 30 min under the same conditions for two

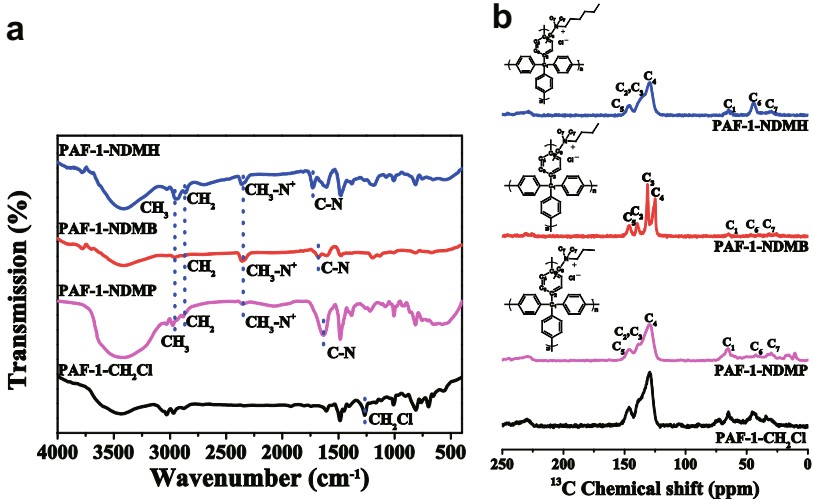

**Fig. 4 Spectroscopic analysis of PAF-1-CH₂Cl, PAF-1-NDMP, PAF-1-NDMB, and PAF-1-NDMH. a** FT-IR spectra of PAF-1-CH₂Cl, PAF-1-NDMP, PAF-1-NDMB, and PAF-1-NDMH. **b** Solid state $^{13}$C NMR of PAF-1-CH₂Cl, PAF-1-NDMP, PAF-1-NDMB, and PAF-1-NDMH.

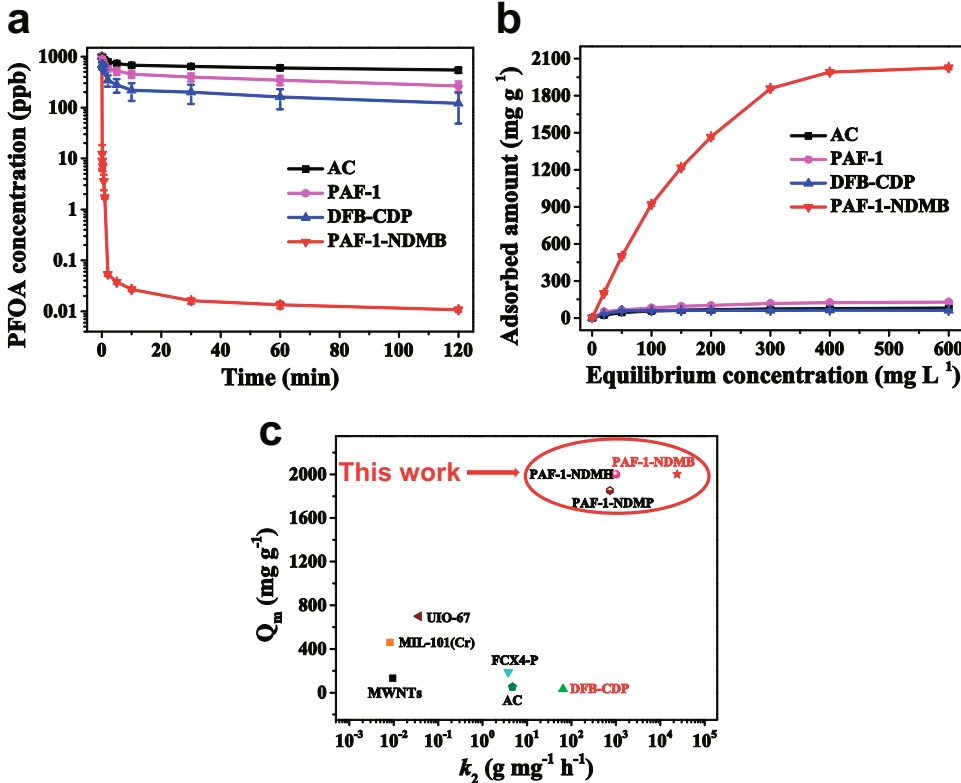

**Fig. 5 Kinetics, adsorption isotherm investigation of PAF-1-NDMB and comparison of PFOA saturation uptake amount and $k_2$ value for PAF-1-NDMB with other benchmark porous materials. a** PFOA sorption kinetics of AC, PAF-1, DFB-CDP, and PAF-1-NDMB. **b** Plotting of equilibrium PFOA adsorption capacity as a function of equilibrium PFOA concentration. **c** The comparison of PFOA saturation uptake amount and $k_2$ value for PAF-1-NDMB with other threshold porous materials, including DFB-CDP[16], MIL-101(Cr)[20], MWNTs[51], FCX4-P[52], UIO-67[53], AC[54]. Red color represents that the materials show PFOA selectivity under the existence of NOM. All the error bars in this figure represent the standard deviation (SD, $n = 3$ independent experiments), data are presented as mean values ± SD.

benchmark sorbents of AC and DFB-CDP and the parent PAF-1, respectively (Fig. 5a, Supplementary Fig. 16). These results thus highlight the 'synergistic binding sites' strategy to construct adsorbents for efficient removal of markedly low concentrations of PFAS. In addition, we evaluated the kinetics performance via both linear[48] and nonlinear[49,50] pseudo second order kinetics models (Supplementary Table 5, Supplementary Figs. 15–18). The higher correlation coefficient ($R^2 = 1$) was obtained via linear pseudo second order kinetics model (Supplementary Table 5). Thus, in this work, the kinetics associated with the adsorption of

PFOA in aqueous solutions using our designed adsorbents were fitted with linear pseudo-second-order kinetic model[48] using Eq. (1):

$$\frac{t}{q_t} = \frac{t}{q_e} + \frac{1}{k_2 q_e^2} \qquad (1)$$

Where $q_t$ (mg g$^{-1}$) is the amount of PFOA adsorbed at time t (min), and $q_e$ (mg g$^{-1}$) is the amount of PFOA adsorbed at equilibrium, and $k_2$ (g mg$^{-1}$ min$^{-1}$) is the pseudo-second order adsorption rate constant. A remarkably high adsorption rate constant ($k_2$) (Fig. 5c) of 24000 g mg$^{-1}$ h$^{-1}$ was obtained for PAF-1-NDMB. To the best of our knowledge, this value is the highest among all adsorbent materials reported so far for PFOA adsorption, such as DFB-CDP (64.8 g mg$^{-1}$ h$^{-1}$)[16], MIL-101(Cr) (0.0082 g mg$^{-1}$ h$^{-1}$)[20], MWNTs (0.00903 g mg$^{-1}$ h$^{-1}$)[51], FCX4-P (3.8 g mg$^{-1}$ h$^{-1}$)[52], UIO-67 (0.036 g mg$^{-1}$ h$^{-1}$)[53], AC (4.72 g mg$^{-1}$ h$^{-1}$)[54].

An important aspect of sorbent's performance metrics is the estimation of the PFOA uptake capacity of PAF-1-NDMB by collecting adsorption isotherm for the removal of PFOA from polluted water as illustrated in Fig. 5b. The obtained data for equilibrium adsorption isotherms were fitted with Langmuir model, giving rise to a high correlation coefficient (R$^2$ = 0.996) (Supplementary Fig. 19). Significantly, the maximum PFOA uptake capacity of PAF-1-NDMB was measured to be 2000 mg g$^{-1}$ (4.8 mmol g$^{-1}$) at the equilibrium concentration of 600 ppm (Fig. 5c), which is 32.0 and 24.1 times higher than that of benchmark materials of DFB-CDP (62.5 mg g$^{-1}$) and AC (83 mg g$^{-1}$), respectively, under the same conditions. This corresponds to the capture of 1.75 PFOA per NDMB group in PAF-1-NDMB (consisting of 0.79 NDMB in 1 g PAF-1-NDMB as illustrated above), thereby implying the great accessibility of PAF-1-NDMB for PFOA. As far as we know, the maximum uptake capacity of PAF-1-NDMB is unprecedented among all adsorbent materials reported thus far for PFOA adsorption, including NU-1000 (507 mg g$^{-1}$)[10], DFB-CDP (33 mg g$^{-1}$)[16], MIL-101 (Cr) (460 mg g$^{-1}$)[20], MWNTs (140 mg g$^{-1}$)[51], FCX4-P (188.7 mg g$^{-1}$)[52], UiO-67 (700 mg g$^{-1}$)[53], AC (52.8 mg g$^{-1}$)[54], all-silica Beta (β) (371 mg g$^{-1}$)[55], resins (1500 mg g$^{-1}$)[23,56]. Although the PAF-1-NDMB exhibited a BET surface area of only 108 m$^2$ g$^{-1}$, it demonstrated a superior uptake capacity due to the efficacious functionalization. Therefore, the obtained results for PAF-1-NDMB set a benchmark for PFOA adsorbent materials.

Moreover, PAF-1-NDMB show high removal efficiency (100%) under different pH (pH = 2–11) conditions, suggesting PAF-1-NDMB exhibited excellent pH tolerance during PFOA adsorption and can be potentially applied in different pH contaminated water (Supplementary Fig. 20). Furthermore, such sorbent can be used to remove other PFAS pollutants such as perfluorooctane sulfonate (PFOS). As a result, PAF-1-NDMB exhibits an exceptionally high PFOS saturation uptake capacity of 2,381 mg g$^{-1}$, extremely fast kinetics with $k_2$ value of 19200 g mg$^{-1}$ h$^{-1}$, which ranks it among the top candidate in benchmarked PFOS sorbents[23,51,53,54,57] (Fig. 6a–c, Supplementary Figs. 21–23). Similarly, it also shows excellent pH tolerance during PFOS adsorption (Supplementary Fig. 24).

**Investigation of PFOA binding interactions**. In principle, such an outstanding PFOA adsorption performance could be mainly attributed to its synergistic capturing effects, resulting from the combination of electrostatic interactions and hydrophobic interactions with PFOA molecules. The excellent binding interactions between PFOA and PAF-1-NDMB were monitored by energy-dispersive X-ray spectroscopy (EDS), zeta potential, IR spectra, and NMR studies. EDS spectra showed the uniform

distribution of fluorine and oxygen atoms from PFOA loading PAF-1-NDMB (PFOA@PAF-1-NDMB), suggesting that PFOA molecules were locked into inner pores of PAF-1-NDMB (Fig. 7a). A fast decrease of zeta potential during the adsorption process was observed from −41.30 mv at 0 s to −47.19 mv at 5 s for PAF-1-NDMB (Supplementary Fig. 25), consistent with the kinetic experiments, thereby suggesting strong electrostatic interactions occurred between PAF-1-NDMB and PFOA molecules. In addition, the IR spectra for PFOA@PAF-1-NDMB (Fig. 7b) exhibited a large chemical shift of C-N stretch mode from 1676 cm$^{-1}$ in PAF-1-NDMB to 1683 cm$^{-1}$ in PFOA@PAF-1-NDMB, attesting the formation of strong electrostatic interactions between COO$^-$ in PFOA and CH$_3$-N$^+$ in PAF-1-NDMB[58,59]. The similar phenomenon can also be achieved in other quaternary ammonium functionalized adsorbents, which show a large shift in IR (5~6 cm$^{-1}$) (Supplementary Table 6, Supplementary Fig. 26). In comparison, PAF-1 that lack of electrostatic binding sites show almost no shift in IR spectra (Supplementary Table 6, Supplementary Fig. 26), which indicated electrostatic interactions play the main role in PFOA adsorption process. Beside the electrostatic interactions, hydrophobic interactions between PFOA and PAF-1-NDMB can also play the important role for improving the performance of PFOA adsorption, which was verified by $^{19}$F MAS NMR spectra for PFOA-loaded PAF-1-NDMB (PFOA@PAF-1-NDMB), which shows a chemical shift of 3 ppm for the peak corresponding to F7 in PFOA (Fig. 7c). The disappearance or weakening of F2~F6 in PFOA@PAF-1-NDMB could be ascribed to the hydrophobic interactions between fluorine chains of PFOA and carbon chains of PAF-1-NDMB[59]. In addition, solid state $^{13}$C NMR spectrum (Fig. 7d) reveals a large shift of 5 ppm for C6 from 45.8 ppm in PAF-1-NDMB to 50.8 ppm in PFOA@PAF-1-NDMB, consistent with the presence of a dramatic hydrophobic interactions between PFOA and PAF-1-NDMB[59]. Moreover, a model pollutant of formic acid (FA) molecule (2.84 × 2.84 × 2.94 Å, a smaller model molecule without hydrophobic chains) shows a very low saturation uptake capacity of 94 mg g$^{-1}$ and small $k_2$ value of 13 g mg$^{-1}$ h$^{-1}$ (Supplementary Figs. 27–31) in PAF-1-NDMB as compared with PFOA molecule, due to the lack of hydrophobic interaction sites compared to PFOA, indicating the important role of hydrophobic interactions in PFAS capture. To further verify the synergistic interactions happened between electrostatic and hydrophobic binding sites in 'synergistic binding sites' sorbents, two mono-functionalized PAF-1 with sole positively charged sites (namely PAF-1-TMA) or hydrophobic sites (namely PAF-1-SE) were synthesized for control experiments (Supplementary Figs. 32–34). Under identical conditions, both PAF-1-TMA and PAF-1-SE exhibited decreased adsorption kinetics and capacities when compared to PAF-1-NDMB with synergistic binding sites, which thus highlight the effect of synergistic binding sites in PFAS adsorption (Fig. 7e, f, Supplementary Figs. 35 and 36). Moreover, PAF-1-TMA performed better in terms of both kinetics and adsorption capacity as compared with PAF-1-SE, indicating a major role of electrostatic interactions during the adsorption process. This could be ascribed to the fact that electrostatic interactions are much stronger than hydrophobic interactions[60].

Furthermore, the density functional theory (DFT) calculations show that the functionalized PAF-1 with both positively charged sites and hydrophobic sites could bind two PFOA molecules via strong coulombic interactions between [(CH$_2$)$_2$N(CH$_3$)$_2$]$^+$ in functionalized PAF-1 and COO$^-$ at PFOA molecules (Fig. 8a–f). The hydrogen bonding between H atoms from the functional group of functionalized PAF-1 and F atoms from PFOA could

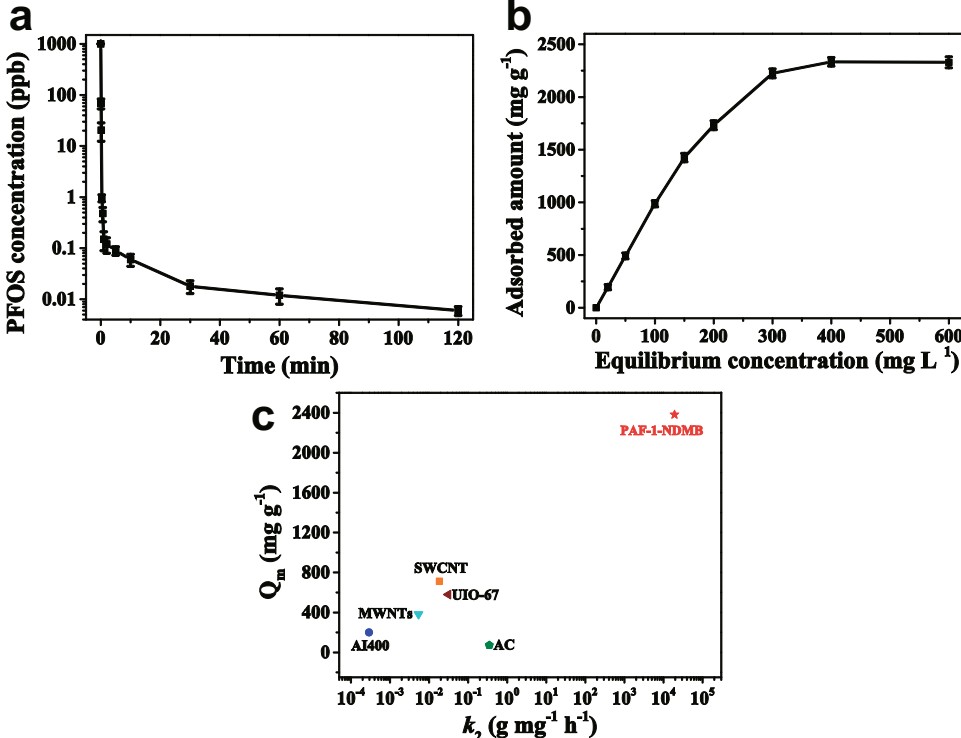

**Fig. 6 Kinetics, adsorption isotherm investigation of PAF-1-NDMB and comparison of PFOS saturation uptake amount and $k_2$ value for PAF-1-NDMB with other benchmark porous materials. a** PFOS sorption kinetics of PAF-1-NDMB. **b** Plotting of equilibrium PFOS adsorption capacity as a function of equilibrium PFOS concentration. **c** The comparison of PFOS saturation uptake amount and $k_2$ value for PAF-1-NDMB with other threshold porous materials, including AI400[23]; MWNTs[51]; UIO-67[53]; AC[54]; SWCNT[57]. All the error bars in this figure represent the standard deviation ($n = 3$ independent experiments), data are presented as mean values ± SD.

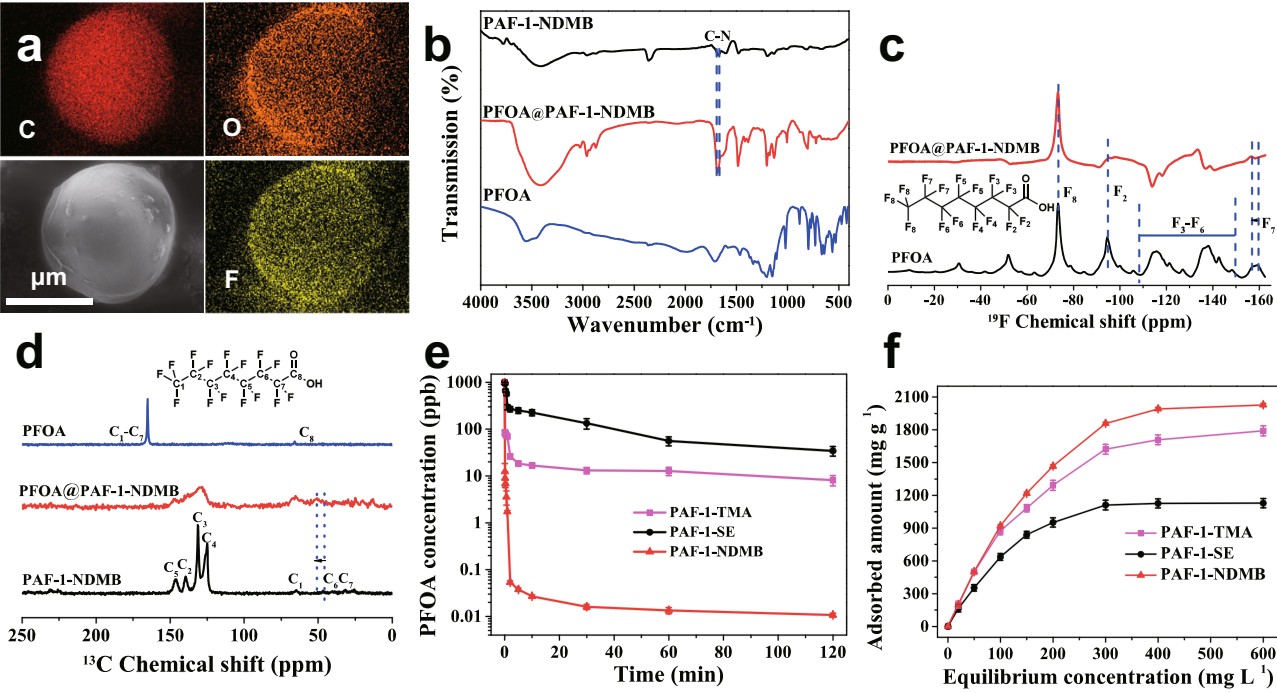

**Fig. 7 Analysis of PFOA binding interactions. a** EDS mapping of PFOA@PAF-1-NDMB. **b** FT-IR spectra of PFOA, PAF-1-NDMB, and PFOA@PAF-1-NDMB. **c** $^{19}F$ NMR of PFOA and PFOA@PAF-1-NDMB. **d** $^{13}C$ NMR spectra of PFOA, PAF-1-NDMB, and PFOA@PAF-1-NDMB. **e** PFOA sorption kinetics of PAF-1-SE, PAF-TMA, and PAF-1-NDMB. **f** Equilibrium PFOA adsorption capacity as a function of equilibrium PFOA concentration. All the error bars in this figure represent the standard deviation ($n = 3$ independent experiments), data are presented as mean values ± SD.

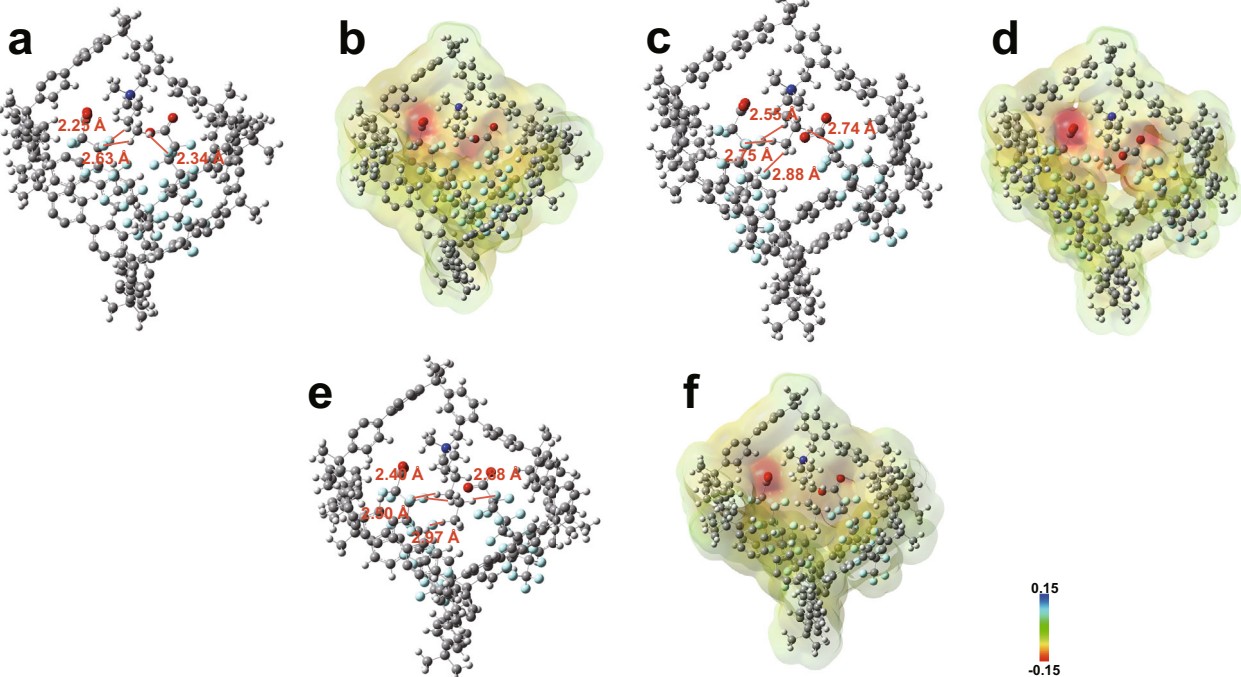

**Fig. 8 DFT calculations analysis.** Map of functionalized PAF-1 with both positively charged sites and hydrophobic sites interacting with PFOA molecules by DFT calculations. **a** Optimized molecular structure of PAF-1-NDMP. **b** Electrostatic potential map of PAF-1-NDMP. **c** Optimized molecular structure of PAF-1-NDMB. **d** Electrostatic potential map of PAF-1-NDMB. **e** Optimized molecular structure of PAF-1-NDMH. **f** Electrostatic potential map of PAF-1-NDMH.

strengthen the molecular interactions between functionalized PAF-1 and PFOA. Moreover, three functionalized PAF-1 show similar binding energy (Supplementary Table 7), and thus an optimized performance of PAF-1-NDMB may due to efficacious functionalization. In all, the synergistic interactions of electrostatic and hydrophobic sites in POPs contributed the high capturing performance for PFAS, thus highlight synergistic capturing sites strategy as a versatile approach for designing PFAS adsorbents.

**Investigation of practical applications**. To investigate the practical applications of our 'synergistic binding sites' adsorbents to withstand organic co-contaminants or natural organic matter (NOM, for example, HA), we performed the breakthrough experiment on PAF-1-NDMB and DFB-CDP in a 500 ppb PFOA solution containing 20 ppm HA (Fig. 9a). The results proved that 300 mg PAF-1-NDMB can treat 3530 ml PFOA (initial concentration with 500 ppb PFOA and 20 ppm HA) containing water, to ensure the residual concentration <53 ppt, lower than the health advisory level (70 ppt) for drinking water set by the EPA of United States, which is 14.6 times than that of DFB-CDP 242 ml (62 ppt). Moreover, the as-synthesized PAF-1-NDMB shows excellent efficiency in removing a heavily contaminated water (400 ppm PFOA solutions), in which 1000 mg PAF-1-NDMB can efficiently purify 9.5 ml PFOA/HA solutions to reach a residual concentration of 31 ppt, which is 19 times higher than that of DFB-CDP (0.5 ml solution, residual concentration of 41 ppt) (Fig. 9b). Besides, in order to confirm that the as-prepared PAF-1-NDMB could be applied to the practical environmental pollutants, we carried out the adsorption experiment to treat the contaminated water from Xiaoqing River, which is the most heavily PFOA polluted water area in China[61,62]. PAF-1-NDMB can remove 99.99% PFOA in practical wastewater within 10 s and reduce PFOA concentrations from initial concentration of 166.27 μg L$^{-1}$ to <70 ng L$^{-1}$ (EPA advisory level) within 10 s (Fig. 9c). Significantly, the absorbent can also be readily regenerated

via washing with the mixed solution of MeOH and saturated NaCl, which can be reused for at least 6 cycles without observable loss of capacities and keep the intact porosity after six cycles (Supplementary Figs. 37–40). Furthermore, the cost of PAF-1-NDMB is 2.5 times lower than benchmark material of DFB-CDP despite the same PFOA adsorption amount, suggesting a potential feasibility toward the application of PFOA capture from contaminated water (Supplementary Table 8). Hence, the results proved that the as-prepared PAF-1-NDMB exhibits excellent ability in the removal of practical contaminated water. The excellent practical performance therefore further proved that 'synergistic binding sites' strategy constructed porous adsorbents can provide a perspective for removing PFOA from contaminated water for industrial applications.

In conclusion, we have described a versatile strategy of constructing PFOA 'synergistic binding sites' sorbent for highly effective and extremely efficient removal of PFOA from contaminated water as demonstrated in the context of a series of quaternary ammonium functionalized POPs, typically PAF-1-NDMB. The constructed PAF-1-NDMB exhibits remarkably high affinity toward PFOA with exceptionally high PFOA uptake capacity over 2000 mg g$^{-1}$, which is 14.8 times enhancement compared with its parent material of PAF-1. And it is 32.0 and 24.1 times higher than benchmark materials of DFB-CDP and AC under same conditions. In view of this finding, it is clearly established that PAF-1-NDMB can effectively reduce the concentration of PFOA from 1000 ppb to an extremely low level of 54 ppt within a few minutes, which is remarkably below the acceptable EPA standards for drinking water (70 ppt). It is worth noting that PAF-1-NDMB exhibits an excellent regeneration ability along with PFOA selectivity under the background of organic co-contaminants. Since such versatile strategy can be readily extended to sorts of POPs, we believe that a large amount of POP-based PFOA sorbents would be emerged in the near future and thus realize the practical application for decontaminating PFAS from polluted water.

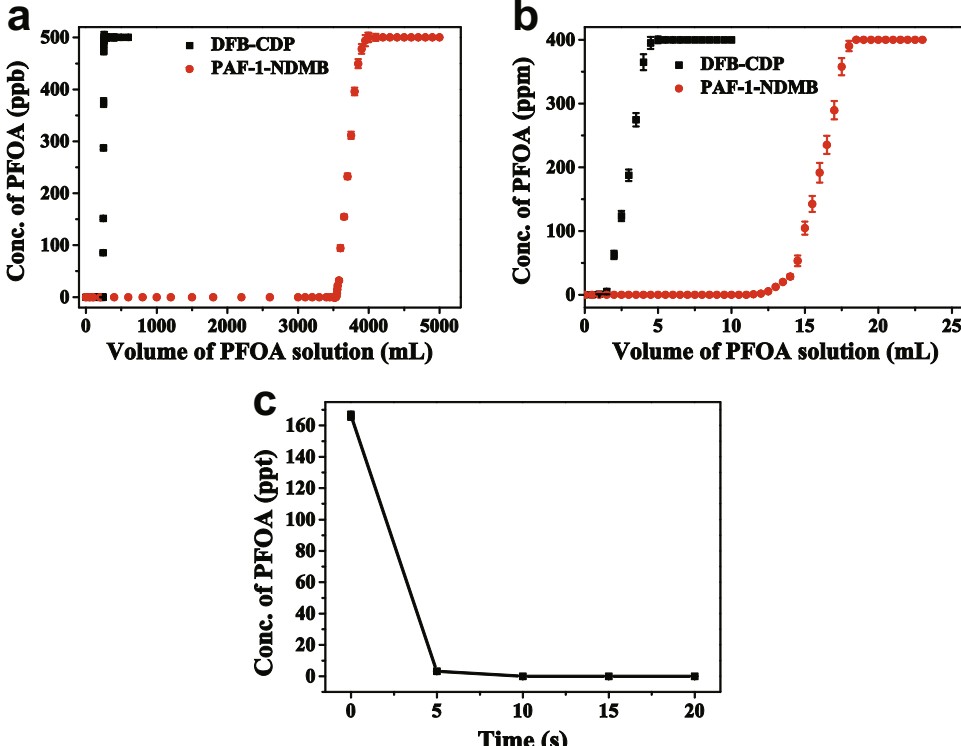

**Fig. 9 Breakthrough experiments and practical environmental contaminated water adsorption experiments.** Breakthrough experiments of (**a**) PAF-1-NDMB and DFB-CDP in aqueous PFOA/HA solutions (PFOA 500 ppb, HA 20 ppm). **b** PAF-1-NDMB and DFB-CDP in aqueous PFOA/HA solutions (PFOA 400 ppm, HA 20 ppm). **c** Adsorption performance of PAF-1-NDMB in removing practical contaminated water (initial concentration with 166.27 ppt). All the error bars in this figure represent the standard deviation ($n = 3$ independent experiments), data are presented as mean values ± SD.

## Methods

**Materials and measurements.** All reagents were purchased from Sigma-Aldrich or Fisher Scientific and used as received, unless otherwise stated. Carbon, hydrogen, and nitrogen elemental analyses were recorded at ambient temperature using Vario EL cube. Infrared spectra were recorded on a TENSOR 37 FTIR spectrometer equipped with an attenuated total reflectance accessory. Solid state $^{13}C$ cross polarization (CP) spectra were collected on a 11 Tesla magnet at a $^{13}C$ frequency of 125.7 MHz under 12 kHz magic-angle spinning (MAS) conditions using Avance III WB 400. Solid-state $^{19}F$ NMR was conducted on a Bruker 600 MHz NMR system with 4.0 mm MAS $^{1}H$-$^{19}F$-X probe. Field emission scanning electron microscopy (FESEM) images were collected on a JSM-7800F with 15.0 kV accelerating voltage. The particle in SEM images was calculated by software of nano measurer 1.2. The particle size distribution of the adsorbents was also analyzed using a dynamic laser light scattering (DLS, DynaPro NanoStar, USA). Energy-dispersive spectroscopy (EDS) images were acquired using FESEM equipped with an X-Max Oxford Instruments XEDS detector (imaged at 10 keV and 4 nA). XPS measurements were performed on an ESCALAB 250 X-ray photoelectron spectroscopy, using Mg Ka X-ray as the excitation source. Gas adsorption measurements were performed with Micromeritics ASAP 2020 plus. The zeta potentials were measured using a Zeta PLAS 190 Plus analyzer (Brookhaven, USA). Quantification of PFOA and PFOS for the adsorption studies was performed by means of Agilent 1200 Series high performance liquid chromatography coupled with ESI-ion trap mass spectrometer Bruker Amazon X (USA). The analytical method was adapted from previously reported literature[16]. Quantification of FA for the adsorption studies was performed by means of ion chromatography (IC, Aquion).

**Synthesis of PAF-1-NDMB.** A 150 mL pressure vessel was charged with PAF-1-CH₂Cl (300 mg), N,N-dimethyl-butylamine (NDMB, 98%, 6.2 g, 0.061 mol), and ethanol absolute (99.5%, 70 ml). The mixture was stirred at 90 °C for three days. After the reaction, the solid was filtered, washed with methanol (98%), and dried under vacuum (100 °C) overnight to afford PAF-1-NDMB, as a light beige powder, yield: 96%. Prior to all measurements, the frameworks were dried under vacuum at 100 °C overnight to remove any residual solvent. Elemental Analysis: C: 58.14%; H: 6.98%; N: 3.84%.

**Synthesis of PAF-1-NDMP.** PAF-NDMP was prepared using the above same procedure by using N,N-dimethylpropylamine (NDMP, 99%) instead of NDMB, yield: 93%. Elemental analysis: C: 64.43%; H: 7.85 %; N: 3.05 %.

**Synthesis of PAF-1-NDMH.** PAF-NDMH was prepared using the above same procedure by using N,N-dimethylhexylamine (NDMH, 98%) instead of NDMB, yield: 95%. Elemental analysis: C: 64.47 %; H: 7.61 %; N: 3.22 %.

*PFOA adsorption experiments.* A 50 mL aqueous of PFOA (1000 ppb), PFOS (1000 ppb), or FA (1000 ppb) was added to an Erlenmeyer flask. Then 20.0 mg sample was added to form a slurry. During the stirring period, the mixture was filtered at intervals (0, 5 s, 10 s, 20 s, 40 s, 1 min, 2 min, 5 min, 10 min, 30 min, 60 min, 120 min) through a 0.2 micron membrane filter for all samples, then the filtrates were analyzed using HPLC-MS and ion chromatography (IC, Aquion) to determine the remaining PFOA, PFOS, or FA content. The experiments were performed for three replicates. The experiments related to other comparing materials including PAF-1, PAF-1-NDMP, PAF-1-NDMH, PAF-1-TMA, PAF-1-SE, DFB-CDP, and coconut shell activated carbon (AC) were conducted based on the same procedure.

The efficiency of pollutant removal by the sorbent was determined by Eq. 2:

$$Pollutant\ removal = \frac{(C_0 - C_t)}{C_0} * 100 \tag{2}$$

where $C_0$ (mg/L) and $C_t$ (mg/L) are the initial and residual concentration of pollutant in the stock solution and filtrate, respectively.

*Isotherm adsorption studies.* Typically, PAF-1-NDMB (10.0 mg) was added to each Erlenmeyer flask containing PFOA, PFOS, or FA solution (100 mL) with different concentrations (20, 50, 100, 150, 200, 300, 400 and 600 ppm). The mixtures were stirred at room temperature for 8 h, and then were filtered separately through a 0.2 micron membrane filter, and the filtrates were analyzed by using HPLC-MS or ion chromatography (IC, Aquion) to determine the remaining PFOA, PFOS, or FA content. The pH of PFOA solution in the kinetic adsorption and isotherm adsorption studies is 6.88. The experiments were performed for three replicates. Other comparing materials including PAF-1, PAF-1-NDMP, PAF-1-NDMH, PAF-1-TMA, PAF-1-SE, DFB-CDP, and coconut shell activated carbon (AC) were conducted based on the same procedure.

Langmuir adsorption fits were generated by Non-linear Least Square Regression in Eq. (3):

$$\frac{1}{q_e} = \frac{1}{q_m} + \frac{1}{C_e * q_m * K_L} \tag{3}$$

Where $q_e$ (mg g$^{-1}$) is the amount of pollutant adsorbed at equilibrium. $q_m$ (mg g$^{-1}$) is the maximum adsorption capacity of adsorbent at equilibrium. $C_e$ (mg L$^{-1}$) is the

residual pollutant concentration at equilibrium. $K_L$ (L mg$^{-1}$) is the Langmuir equilibrium constant.

*Density functional theory calculations.* Density functional theory (DFT) calculations were performed to optimize the ground-state geometry of PAF-1-NDMP$^+$/PFOA$^-$, PAF-1-NDMB$^+$/PFOA$^-$, as well as PAF-1-NDMH$^+$/PFOA$^-$ and obtain the corresponding electrostatic potential map using the long-range dispersion corrected functional CAM-B3LYP together with the basis set 6-31 G (d,p) as implemented in Gaussian 09 program (D. 01).

*Zeta potential measurements.* A 50 mL aqueous of PFOA (1000 ppb) was added to an Erlenmeyer flask. Then 20.0 mg PAF-1-NDMB was added to form a slurry. During the different stirring period (0, 5 s, 10 s, 20 s, 40 s, 1 min, 2 min, 4 min), the zeta potentials were collected using a Zeta PLAS 190 Plus analyzer (Brookhaven, USA). The experiments were performed for 3 replicates.

*Breakthrough experiments.* The breakthrough experiments were performed under a PFOA concentration of 500 ppb or 400 ppm by using PAF-1-NDMB or DFB-CDP as adsorbents. The experimental device was illustrated in Supplementary Fig. 41. 6 L PFOA solutions (500 ppb or 400 ppm) were mixed with 20 mg L$^{-1}$ humid acid and stirred for 12 h. Firstly, 300 mg of PAF-1-NDMB and DFB-CDP sample was packed into a pipette (inner diameter = 3.3 mm) to form an adsorption column and the obtained packed sample length was about 7.8 cm. The simulated contaminated water was pumped into the column and the filtrates were collected at intervals and filtered with a 0.2 μm Whatman inorganic syringe filter. The filtrates were analyzed using HPLC-MS to determine the contents. The experiments were performed for three replicates.

*Practical environmental contaminated water adsorption experiments.* We have collected the practical PFAS polluted water from Xiaoqing River of China, which was heavily polluted by PFOA[61,62]. The pretreatment of the water samples were according to the previous report[61,62]. The initial concentration of the collected practical contaminated water (pH = 6.58) is 166.27 μg L$^{-1}$ based on our experimental results. A 50 mL polluted water was then performed under a same procedure of kinetic adsorption studies by using 20 mg PAF-1-NDMB as sorbent. The filtrates were analyzed using HPLC-MS to determine the contents. The experiments were performed for 3 replicates.

## Data availability
The $^1$H NMR, $^{13}$C NMR and HRMS data for tetraphenylmethane and tetrakis(4-bromophenyl)methane in this study available at https://figshare.com/s/e34c5be7b2a2ef6e8a45. Other data that support the findings of this study are available within the article and supplementary information files, or available from the corresponding authors on request.

## Code availability
Gaussian 09 (Revision D.01) for the DFT calculations is available at https://gaussian.com/glossary/g09/.

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

## Acknowledgements

The authors acknowledge National Science Foundation of China (NO. 21978138 and 22035003) and the Fundamental Research Funds for the Central Universities (Nankai University) for financial support of this work. Financial support was also provided by the Haihe Laboratory of Sustainable Chemical Transformations. Partial support from the U.S. National Science Foundation (CBET-1706025) and the Robert A. Welch Foundation (B-0027) (SM) as well as from Researchers Supporting Program project no (RSP-2022/79) at King Saud University, Riyadh, Saudi Arabia (AN) is also acknowledged.

## Author contributions

B.L., X.-H. B., and S.M. conceived and designed the research. X.L. and B.L. co-wrote the manuscript. X.L. and B.L. carried out the materials design, synthesis and characterization as well as performed PFOA sorption experiments and data analysis. J.Y. carried out the DFT calculations. C.Z., J.L., Z.Z., J.L., F.S., Z.Y., Z.S., and A.N. contributed to materials characterization. All authors discussed the results and commented on the manuscript.

## Competing interests

A patent has been applied for by Nankai University with B.L. and X.L. as named inventors. The patent application number is CN202110416181.6. The remaining authors declare no competing interests.
