## [Peer review file · Nature Communications]

REVIEWER COMMENTS

Reviewer #1 (Remarks to the Author):

The synthesis and characterization of a novel porous organic polymer equipped with a combination of electrostatic and hydrophobic functionality for PFOA adsorption is reported in this manuscript. The rational material design aspects and record adsorption capacities/kinetics are impressive. All characterizations have been performed to very high standards and discussions are valid to support the conclusions. In my opinion, the results present in this manuscript meet the high standards set by Nature Communications. I suggest a few minor points which should be addressed before acceptance of the manuscripts. The listed comments may be considered to further improve the quality of the article.

1. Typo error, page 16, line 334, repetition of the word “breakthrough”.
2. What is the role of humic acid in the adsorption of PFOA? This need to be mentioned in the manuscript.
3. The XPS survey spectra are missing.
4. It seems that the N,N-dimethyl-butylamine based PAF-1 (PAF-1-NDMB) showed much better performance of PFAS adsorption compared to hexyl and propyl derivatives. However, the surface area of PAF-1-NDMB is less (108 m²g⁻¹) compare to the propyl derivative (602 m²g⁻¹). Although the role of strong hydrophobic interaction of PAF-1-NDMB with PFAS is mentioned in the manuscript, the computational data of PAF-1-NDMP and PAF-1-NDMH are missing.
5. The average particle size of the adsorbents (PAF-1-NDMP, PAF-1-NDMB and PAF-1-NDMH) is missing. The SEM histogram and DLS of these compounds should be provided which will be highly useful to see whether any role of particle size on the PFAS adsorption is present or not.
6. Some new relevant literature could be cited, such as Environ. Sci. Technol. 2021, 55, 22, 15162–15171; Journal of Hazardous Materials Letters, 2021, 2, 100034; Angew. Chem. Int. Ed. 2020, 132, 14190-14194; Chemistry of Materials, 2021, 33, 3276-3285.

Reviewer #2 (Remarks to the Author):

[please also see attached file which includes remarks made by reviewer #2]

1. These adsorption results are outstanding, surpassing the benchmark PFOA adsorbent materials of DFB-CDP and commercial AC. However, it is not clear whether the authors synthesized the DFB-CDP again for their work or they just extracted the adsorption data published by the original DFB-CDP authors (reference #16 in this manuscript). The type of studied AC was also not explicitly mentioned. This information is important for cross-comparison with other reports.
2. Although the written sentences could convey the planned meanings, the overall grammar structure for the manuscript needs a minor polish. Therefore, it would be best if the authors can send the manuscript to a professional editor before resubmitting to this journal.
3. The principles behind efficient PFOA adsorption are acceptable, comprising electrostatic interactions and high adsorbent's hydrophobicity. However, the latter element is inconsistent. Sometimes, the authors claimed hydrophilicity is preferred while in other instances, a hydrophobic adsorbent is better. The schematics in Fig. 3 should be corrected to present the correct proposed reaction mechanisms.
4. In summary, this manuscript can be accepted for publication in Nature Communications, subject to minor revisions, some of which are included in annotated manuscript attached.

Dr. Shengqian Ma
Professor
Robert A. Welch Chair in Chemistry

Department of Chemistry
University of North Texas
1508 W Mulberry St
Denton, TX 76201
E-mail: Shengqian.Ma@unt.edu
Phone: 940-369-7137
Fax: 940-565-4318
Website: <http://www.chemistry.unt.edu/~sqma/>

Thank you very much for the opportunity to revise our manuscript entitled “***Installation of Synergistic Binding Sites onto Porous Organic Polymers for Efficient Removal of Perfluorooctanoic Acid***” (Manuscript ID: NCOMMS-21-49438). We really appreciate the comments and suggestions from the reviewers, and we have revised our manuscript accordingly as detailed in the responses below. The corresponding changes have been highlighted in yellow in the main text and Supplementary Information. In addition, we have also revised the format of the manuscript according to the requirement of *Nature Communications*. We hope this revision will be satisfactory for publication in *Nature Communications*. We thank you again for the smooth handling of our manuscript.

Reviewer: 1

Comment 1: Typo error, page 16, line 334, repetition of the word “breakthrough”.

Response 1: Thanks for pointing out this typo. It has been fixed.

Comment 2: What is the role of humic acid in the adsorption of PFOA? This need to be mentioned in the manuscript.

Response 2: Thanks for the constructive suggestion. The humic acid (HA) served as a model of organic co-contaminants or natural organic matters (NOM) to evaluate the sorption performance of our “synergistic binding sites” adsorbents in practical applications (*J. Am. Chem. Soc.* 2017, 139, 7689-7692). This explanation has been added in the revised manuscript (please see the details in page 4, and Fig. 1).

Comment 3: The XPS survey spectra are missing.

Response 3: Thanks for pointing this out. The XPS survey spectra have been added as Supplementary Fig. 1. in the revised Supplementary Information.

Comment 4: It seems that the N,N-dimethyl-butylamine based PAF-1 (PAF-1-NDMB) showed much better performance of PFAS adsorption compared to hexyl and propyl derivatives. However, the surface area of PAF-1-NDMB is less ($108 \text{ m}^2\text{g}^{-1}$) compare to the propyl derivative ($602 \text{ m}^2\text{g}^{-1}$).

Although the role of strong hydrophobic interaction of PAF-1-NDMB with PFAS is mentioned in the manuscript, the computational data of PAF-1-NDMP and PAF-1-NDMH are missing.

Response 4: Thanks for the constructive suggestion. Per the suggestion, the computational data of PAF-1-NDMP and PAF-1-NDMH have been added in the revised manuscript (please see page 12, Fig. 8 and Supplementary Table 7 for details).

Comment 5: The average particle size of the adsorbents (PAF-1-NDMP, PAF-1-NDMB and PAF-1-NDMH) is missing. The SEM histogram and DLS of these compounds should be provided which will be highly useful to see whether any role of particle size on the PFAS adsorption is present or not.

Response 5: Thanks for the constructive suggestion. Per the suggestion, the data of SEM histogram and dynamic laser light scattering (DLS) have been added in the revised manuscript (please see page 6, Supplementary Figs. 3-4 for details).

Comment 6: Some new relevant literature could be cited, such as Environ. Sci. Technol. 2021, 55, 22, 15162–15171; Journal of Hazardous Materials Letters, 2021, 2, 100034; Angew. Chem. Int. Ed. 2020, 132, 14190-14194; Chemistry of Materials, 2021, 33, 3276-3285.

Response 6: Thanks for the suggestion. These relevant literatures have been cited as Ref. 2, 4, 10, 55 in the revised manuscript.

Reviewer: 2

Comment 1: These adsorption results are outstanding, surpassing the benchmark PFOA adsorbent materials of DFB-CDP and commercial AC. However, it is not clear whether the authors synthesized the DFB-CDP again for their work or they just extracted the adsorption data published by the original DFB-CDP authors (reference #16 in this manuscript). The type of studied AC was also not explicitly mentioned. This information is important for cross-comparison with other reports.

Response 1: Thanks for your suggestion. We synthesized the DFB-CDP according to the procedure of reference (*J. Am. Chem. Soc.* 2017, 139, 7689-7692), and the synthetic procedures of DFB-CDP have been added in the revised Supplementary Information (please see page S4 for details). All the comparing data of DFB-CDP were obtained *via* sorption experiments as mentioned in our manuscript (please see page 16-17 for details). The type of AC was mentioned in page 5 in the revised manuscript.

Comment 2: Although the written sentences could convey the planned meanings, the overall grammar structure for the manuscript needs a minor polish. Therefore, it would be best if the authors can send the manuscript to a professional editor before resubmitting to this journal.

Response 2: Thanks for the advice. The manuscript has been polished by some professional editor to fix all possible grammatical errors.

Comment 3: The principles behind efficient PFOA adsorption are acceptable, comprising electrostatic interactions and high adsorbent's hydrophobicity. However, the latter element is inconsistent. Sometimes, the authors claimed hydrophilicity is preferred while in other instances, a hydrophobic adsorbent is better. The schematics in Fig. 3 should be corrected to present the correct proposed reaction mechanisms.

Response 3: Thanks for the constructive suggestion. Since the PFOA is often present in aqueous solutions, the macroscopic particle of adsorbent should show certain hydrophilicity

and wettability to ensure the mass transfer of PFOA toward adsorbent in water. The introduced ionic quaternary ammonium groups would increase hydrophilicity of functionalized PAF-1 particles compared to parent PAF-1 with a hydrophobic property. However, at the molecular level, the hydrophobic carbon chain was introduced into the adsorbent to strengthen the adsorptive affinity of PFOA molecule with a hydrophobic fluorine chain toward adsorbent via hydrophobic interactions. We have added relevant discussions in the revised manuscript (please see page 3-5 and page 7 in the manuscript), and Fig. 3 has also been revised accordingly.

Comment 4: In summary, this manuscript can be accepted for publication in Nature Communications, subject to minor revisions, some of which are included in annotated manuscript attached.

Response 4: We thank the reviewer for taking time to review our manuscript and the support for our work. We have revised the manuscript according to the suggestions/comments from the reviewer.

Comment 5: References 6 to 9 are slightly outdated. It would be better to also include recent publications circa 2018 - 2022.

Response 5: Thanks for the suggestion. We have updated references with some recent publications added.

Comment 6: Please include the range of capacity reported in literature. For example, from reference 11 to 21 (600 - 1000 mg/g). Please include the range of equilibrium adsorption time.

Response 6: Thanks for the suggestion. Per the suggestion, we have included the range of capacity and the range of equilibrium adsorption time (please see page 3 in the revised manuscript).

Comment 7: The authors need to be consistent either to use Porous Organic Polymers (POPs) or Porous Aromatic Frameworks (PAFs).

Response 7: Thanks for the suggestion. We have defined the description of Porous Organic Polymers in the revised manuscript.

Comment 8: What is meant by densely capturing sites? Porous, or functionalized? The desired nature of the adsorption sites should be specified.

Response 8: Thanks for the suggestion. Per the suggestion, we have specified the desired nature of the adsorption sites in the revised manuscript.

Comment 9: What structures? The internal adsorbent framework or the external surface structure? The word "structures" should be specified.

Response 9: Thanks for the suggestion. Per the suggestion, the word "structures" have been specified in the revised manuscript.

Comment 10: Please refer this chloromethylation procedure to Fig. 2. Please also explain why chloromethylation is necessary prior to performing the functionalization? Is this the best method over other alternatives?

Response 10: Thanks for the comments. Chloromethylation is a common method for the post-synthetic modifications of POPs due to the high reactivity of chloromethyl group with amines. Some examples have proved the excellent feasibility to introduce amine groups via such reaction. For example: *Angew. Chem., Int. Ed.*, 51, 7480-7484 (2012); *Chem. Sci.*, 7, 2138-2144 (2016). We have added the corresponding discussion in the revised manuscript (please see page 5 for details).

Comment 11: The authors need to discern between the Cl⁻ coming from the ion exchange process and the converted Cl⁻ from the excess NDMP, NDMB and NDMH conversion as stated in line 107. The Cl⁻ values from the modified adsorbents should be compared after their functionalization and after they were used for PFOA adsorption, respectively.

Response 11: Thanks for the insightful comment. We have added the corresponding discussion in the revised manuscript (please see page 7 for details).

Comment 12: The authors should also include calculated data from a nonlinear pseudo second order kinetics model. Please refer to Simonin (2016) and Kajjumba et al. (2019) as they have shown that a nonlinear estimation is preferred than using a linearized model. It might be because a linear pseudo second order model was used, hence, why the calculated kinetics coefficient is extremely high.

Response 12: Thanks for the constructive suggestion. Per the suggestion, we evaluated the kinetics performance *via* both linear and nonlinear pseudo second order kinetics models (Supplementary Table 5, Supplementary Figs. 15-18). The k_2 value of PAF-1-NDMB obtained based on linear pseudo second order kinetics model is similar as the results based on nonlinear pseudo second order kinetics model (Supplementary Table 5). While the k_2 values of PAF-1-NDMP and PAF-1-NDMH obtained based on the nonlinear pseudo second order kinetics model are much higher than the results based on the linear pseudo second order kinetics model. Although a similar or higher k_2 value can be achieved by nonlinear pseudo second order kinetics model, the higher correlation coefficient (R^2 values) was obtained via linear pseudo second order kinetics model compared to nonlinear pseudo second order kinetics model. Thus, in this work, the former was used to evaluate the k_2 value (please see page 8).

Comment 13: How did the authors calculate this (This corresponds to the capture of 1.75 PFOA per NDMB group in PAF-1-NDMB)? A clear correlation must be stated originating from their statement on line 103.

Response 13: Thanks for the suggestion. We have added the calculation details in the revised manuscript (please see page 9 for details).

Comment 14: No standard error bars were included in the kinetics experiment (Fig 5a).

Response 14: Thanks for the constructive comments. The standard error bars in the kinetics experiment have been added (please see Fig. 5a for details).

Comment 15: The authors can highlight the fact that PAF-1-NDMB only has 108 m²/g of surface area, yet with superior uptake capacity due to the efficacious functionalization.

Response 15: Thanks for the comment. We have added the related discussion in the revised manuscript (please see page 9 for details).

Comment 16: This term "neglect shift" is not understood.

Response 16: Thanks for pointing this out. It has been revised as "almost no shift" in the revised manuscript.

Comment 17: Please include the molecular dimension of formic acid and discuss this dimensional difference aspect compared to PFOA too.

Response 17: Thanks for the suggestion. We have added the molecular dimension of formic acid and discuss the difference compared to PFOA in the revised manuscript (please see page 11, Supplementary Fig. 27 for details).

Comment 18: Please include the schematic in the Supplementary Information to understand the final modified structure. Same comment goes to PAF-1-TMA.

Response 18: Thanks for the suggestion. We have added the schematic in the revised Supplementary Information (please see Supplementary Fig. 32. for details).

Comment 19: It is understood that the cationic PAF-1-TMA and the hydrophobic PAF-1-SE showed reduced adsorption kinetics and capacity, but these discrepancies are not explained in detail. Please explain which factor is more important between the two and provide your justification.

Response 19: Thanks for the insightful comments. The related discussions have been added in the revised manuscript (please see page 11-12 for details).

Comment 20: Why is the adsorbent ratio is not the same with earlier experiments of 0.4 mg adsorbent/mL PFOA. This time it is much lower, approximately 0.085. Please justify the choice of this adsorbent/adsorbate ratio.

Response 20: Thanks for the comments. For the kinetics experiment, the volume of PFOA solutions is fixed, and we can control the ratio of adsorbent/adsorbate. While in the breakthrough experiment, the PFOA solution was dynamically passed through the adsorption bed, and the adsorption performance was determined by dynamically monitoring the final collected volume. Therefore, the adsorbent/adsorbate ratio is not fixed in breakthrough experiments.

Comment 21: Details for the DFT calculations were not included in the Methods section. Please include this important information accordingly.

Response 21: Thanks for the suggestion. We have added the details of DFT calculations to the methods section in the revised manuscript (please see page 17 for details).

Comment 22: Please indicate the specific SEM images acquired from these two distinct instruments to avoid confusion.

Response 22: Thanks for the suggestion. We have revised the related descriptions for SEM images in the revised manuscript (please see page 14 for details).

Comment 23: Please state the vacuum drying temperature.

Response 23: Thanks for the suggestion. The vacuum drying temperature has been stated in the revised manuscript (please see page 15 for details).

Comment 24: As electrostatic interactions are one of the dominant PFOA adsorption mechanisms, please explain in length about the sample preparation for zeta potential measurements.

Response 24: Thanks for the suggestion. We have added the sample preparation for zeta potential measurements in the methods section of the revised manuscript (please see page 17 for details).

Comment 25: Please amend the schematic as the R groups should have substituted the Cl⁻ from the earlier grafted CH₂-Cl on the PAF-1 (Fig. 3). Therefore, the scheme should not include the Cl⁻ anymore. Second, it should be the cationic N⁺ that is attached to the original Cl⁻ position on the arene ring. It should not be the whole NDMP/NDMB/NDMH that is grafted onto the middle of the benzene ring. This illustration can be misleading.

Response 25: Thanks for the suggestion. Per the suggestion, we have revised the Fig. 3 in the revised manuscript.

Comment 26: Please change the y-axis title for Fig.6(a) as PFOS concentration not PFOA. Please also add the label to Fig. 6 (a) and (b) as only Fig. 6 (c) is labeled.

Response 26: Thanks for the suggestion. We have corrected the errors in Fig. 6.

Comment 27: It would be reasonable to include another figure as Fig. 9(c) to show the adsorption data from the real river water sample. This information is an excellent piece of information worthwhile mentioning.

Response 27: Thanks for the constructive suggestion. We have revised the Fig. 9 to include another figure to show the adsorption data from the real river water sample.

Again, we thank the reviewers for the constructive comments and suggestions, which have made our manuscript much improved.

Sincerely,

Shengqian Ma, PhD

Professor and Welch Chair in Chemistry

REVIEWERS' COMMENTS

Reviewer #1 (Remarks to the Author):

All of my comments have been satisfactorily addressed. No further changes needed.

Reviewer #2 (Remarks to the Author):

The authors have corrected the manuscript in line with our suggestions, and the paper should proceed to publication.